# Single Ferritin Nanocages Expressing SARS-CoV-2 Spike Variants to Receptor and Antibodies

**DOI:** 10.3390/vaccines12050446

**Published:** 2024-04-23

**Authors:** Monikaben Padariya, Umesh Kalathiya

**Affiliations:** International Centre for Cancer Vaccine Science, University of Gdansk, ul. Kładki 24, 80-822 Gdansk, Poland

**Keywords:** antibodies, spike, host cell receptor, mutations, COVID-19, nanocages, ferritin, SARS-CoV-2

## Abstract

SARS-CoV-2 virus variants of concern (VOCs) have rapidly changed their transmissibility and pathogenicity primarily through mutations in the structural proteins. Herein, we present molecular details with dynamics of the ferritin nanocages stitched with synthetic chimeras displaying the Spike receptor binding domains (RBDs). Our findings demonstrated the potential usage of ferritin-based vaccines that may effectively inhibit viral entry by blocking the Spike–ACE2 network and may induce cross-protective antibody responses. Taking the nanocage constructs into consideration, we evaluated the effects of variants on the docked interface of the SARS-CoV-2 Spike RBD with the ACE2 (angiotensin-converting enzyme 2) host cell receptor and neutralizing antibodies (Abs). Investigating the VOCs revealed that most of the mutations reported a possibly reduced structural stability within the Spike RBD domain. Point mutations have moderate or no effect for VVH-72, CR3022, and S309 Abs when bound with the Spike RBD, whereas a significant effect was observed for B38, CB6, and m396 over the surface of the H-ferritin nanocage. In addition to providing useful therapeutic approaches against COVID-19 (coronavirus disease 2019), these structural details can also be used to fight future coronavirus outbreaks.

## 1. Introduction

Ferritin nanocages are known for their inherent cavity that varies in size, biocompatibility, and high water solubility, enabling potential applications in diverse disciplines such as medicine, food, nutrition, etc. Ferritin nanocages can vary in size (16-mer, 24-mer (a typical size), 48-mer, etc.), and among them, H-ferritins are well known for use in several applications [1,2]. The monodispersed architectures of these self-assembled ferritin nanocages generated by a small number of units have attracted researchers and are versatile nanoparticles among many types of nanocarriers such as liposomes, cyclodextrin, etc. [3,4,5,6,7]. The usage of ferritin molecules has been extensive for vaccine generation as they can be easily degraded and also have low toxicity and immunogenicity [8]. In oncology, these H-ferritin nanocages can also be used as a delivery system of indocyanine green (ICG) for fluorescence-guided surgery applications [9], alongside drug delivery in cancer therapies [10]. In addition to the drug molecules encapsulated within the nanocages, ferritins are often combined with different modifications to enhance their therapeutic effects [8]. They have been reported as an attractive molecule for vaccine development by displaying antigen molecules over its surface (in the active form) that could enhance immune response or modulating binding with the target gene [11,12,13,14,15].

Among the different SARS-CoV-2 structural proteins, the virus uses the glycosylated Spike to directly contact the host cell receptor through the ACE2 (angiotensin-converting enzyme 2). Individual subunits from the Spike homotrimer (a functional form) were found to gain different pre- and post-fusion conformations. In particular, the pre-fusion state or conformation of the Spike protein has been widely used to guide different therapeutics. In our previous work [16], we investigated the dynamics of different ferritin (H and L) nanocages that can represent receptor binding domains (RBDs) from the SARS-CoV-2 Spike protein over its surface. Investigating differently sized linkers stitching ferritin and Spike RBD highlighted that a five amino acid (GGGGS) linker can be an optimal length to maintain the ‘up’ active conformations of RBDs [16,17]. These Spike RBDs forming direct interactions with the ACE2 host cell receptor [18,19,20] are potential therapeutic targets against SARS-CoV-2 infection. The glycosylated homotrimer structure of the Spike protein has S1 and S2 subunits [21]; and the Spike RBD domain (329–521 amino acids; aa) belongs to the S1 subunits. Antibodies designed especially for these Spike RBDs can efficiently neutralize the infection [20]. The longevity of antibody response against the SARS-CoV-2 Spike protein from a dataset of vaccinated candidates revealed that it can be sustained up to 12 months [22]. Explicitly, we evaluated the behavior of the Spike RBD over the ferritin nanocages and their binding regions responsible for interacting with the host ACE2 receptor and different antibodies (Abs) such as VVH-72, B38, CB6, CR3066, m396, and S309 (Figure 1) [23,24,25]. These antibodies have a slight different interaction network or binding mechanism with the SARS-CoV-2 Spike RBD and ACE2 receptor [26,27,28,29,30]: (i) the VVH-72 and B38 Abs dock the Spike RBD, which eventually prevents its interaction with ACE2, (ii) m296 forms interactions with the Spike RBD without mimicking ACE2 (however, it can create allosteric effects that can block it with ACE2), (iii) CB6 mimics ACE2 binding to the Spike RBD, and (iv) S309 and CR3022 Abs interact with the Spike RBD domain without disrupting its interaction with ACE2. Alongside investigating dynamics and interaction patterns or patches of the Spike RBD over ferritin nanocages, using in silico techniques (Molecular Operating Environment, MOE; Chemical Computing Group Inc., Montreal, QC, Canada) [31], we studied its change in the binding affinity toward ACE2 or antibodies with respect to different mutations (variants of concerns; VOC) derived from the SARS-CoV-2 variants (Alpha, B.1.1.7; Beta, B.1.351; Delta, B.1.617.2 and B.1.617; Gamma, P.1; Lambda, C.37; and Omicron, B.1.1.529 (Figure 1) [32,33].

## 2. Materials and Methods

### 2.1. The Model Selection and Chimera Constructs of Ferritin–RBD–Receptor or Antibody Complexes

The constructs of H-ferritin connected by different sized linkers with the Spike RBD were designed in our previous work [16], which were generated based on the available structures from the PDB database (protein data bank; www.rcsb.org). The crystal structure of the Spike RBD domain (PDB ID: 6lzg [34]) in the ’up’ active conformation docked with the ACE2 receptor was considered in this study. Implementing the nanocage generation module from the PDBePISA (proteins, interfaces, structures, and assemblies) [35], the 24-mer ferritin nanocage structure was built and was further optimized using molecular dynamics simulations (MDSs) [16]. The optimal five amino acids GGGGS linker that stabilizes the orientation in the ‘up’ active conformation of the Spike RBD domains over the H-ferritin nanocages was taken into consideration in our study. These ferritin–RBD–ACE2 or antibody systems were designed using different homology modeling pipelines and modules within the MOE package [16,18].

To investigate the binding or the docked interfaces of antibodies with the SARS-CoV-2 Spike RBD domain over the H-ferritin nanocages, the following structures were considered: (i) and (ii) VVH-72 (PDB ID: 6waq [30]) and B38 (PDB ID: 7bz5 [36]) forms an interaction with the Spike RDB where the ACE2 binds, (iii) m396 binds to the Spike RBD, preventing Spike–ACE2 docking (PDB ID: 2dd8 [26]), (iv) CB6 mimics the ACE2 interaction site over the Spike RBD (PDB ID: 7c01 [27]), (v) S309 (PDB ID: 6wps [28]) and (vi) CR3022 (PDB ID: 6w41 [29]) enable binding with the Spike RBD domain without disrupting its interaction with ACE2 [24]. The conformation or position of all six antibodies were sustained as retrieved from the PDB database, which were further considered to design homology models of the ferritin–linker–RBD (PDB ID: 6lzg [34]) complexes within the MOE package. Implementing the CHARMM27 forcefield [37], the constructed model systems over the ferritin–RBD–antibody were energy minimized. The MOE and BIOVIA Discovery Studio (DassaultSystèmes, BIOVIA Corp., San Diego, CA, USA) packages were used to represent proteins, antibodies, or nanocage structures.

### 2.2. The ‘Residue Scan’ Approach to Identify Change in the Binding Affinity

These ferritin–linker–RBD–receptor or antibody modeled structures were used to identify changes in the binding affinity or stability upon inserting point mutations retrieved from different SARS-CoV-2 variants: Alpha (B.1.1.7), Beta (B.1.351), Delta (B.1.617.2 and B.1.617), Gamma (P.1), Lambda (C.37), and Omicron (B.1.1.529) [32,33]. An optimized “residue scan” pipeline or module incorporated within the MOE packages was used to generate mutant models. Applying the concept of relative binding free energy (∆∆G_bind_; thermostability), differences of wild-type and mutant energies were retrieved using the Boltzmann relative stability of the ensemble. Apart from the Spike RBD mutations that emerged from different SARS-CoV-2 variants, we further evaluated them with our pipelines, where a single residue can be mutated at a time with all possible amino acids: A, R, N, D, C, Q, E, G, H, I, L, K, M, F, P, S, T, W, Y, and V. For these mutations, the change in the binding affinity (∆∆G) of the Spike RBD–receptor or antibodies over the ferritin nanocage surface was measured using the LowModeMD ensemble (10k iterations) and CHARMM27 forcefield [16,38]. The negative (∆∆G) structural stability or binding affinity for a mutation induces the stability within the structure or protein–antibody energy, whereas a positive value shows contradictory results.

## 3. Results

Several studies have shown that the ferritin nanocages can explicitly display proteins or peptides over its surface [39,40,41,42,43,44,45,46]. In particular, the Spike RBD targeting vaccines and neutralizing antibodies can effectively block the viral entry by interrupting the binding between the ACE2 host cell receptor and SARS-CoV-2 Spike RBD domain [41,42,43,44,45,46]. Since this could provide useful therapeutic approaches against COVID-19 (coronavirus disease 2019), we investigated the mutational landscape of the Spike RBD with the ACE2 host–cell receptor or different antibodies displayed over the ferritin nanocages.

### 3.1. Influence of Mutational Landscape over the Spike RBD–ACE2 Interface

Mutations within the Spike RBD regions lead or have been correlated to the emergence of different SARS-CoV-2 variants such as Alpha (B.1.1.7), Beta (B.1.351), Delta (B.1.617.2 and B.1.617), Gamma (P.1), Lambda (C.37), and Omicron (B.1.1.529; Figure 2). Since these variants enhance the binding and infectivity of the virus compared to the wild type, we performed an analysis of their mutations by introducing them in an optimized Spike RBD over the H-ferritin (24 subunits) nanocage structures (Figure 2 and Appendix A). These mutated SARS-CoV-2 Spike RBDs over the ferritin nanocage stitched by a five amino acids linker were reconstructed with the ACE2 receptor [34,47]. A comparative Spike RBD sequence variability from different SARS-CoV-2 variants guided the location of point mutations. The Spike RBD from the SARS-CoV-2 Omicron variant contains the highest number of mutations (15 amino acid changes) compared to the SARS-CoV-2 α, β, γ, and δ variants.

Such substitutions within the Spike RBD domain are crucial for SARS-CoV-2 infectivity, since they severely affect the binding of the Spike protein with the ACE2 host cell receptor. Subsequently, the potential influence of these mutations on properties like protein stability and binding affinity have been evaluated for the ferritin–RBD–ACE2 complexes (Figure 2 and Appendix A). Replacing a specific mutated residue with every possible substitution (A, R, N, D, C, Q, E, G, H, I, L, K, M, F, P, S, T, W, Y, and V), the majority of the exchanges were reported to reduce the stability (positive values) of the Spike RBD domain (Appendix A). A set of mutations at positions 496, 498, and 501 were among the highest inducing (negative values) stability of the Spike RBD domain (Appendix A). Investigating residues involved in the Spike RBD–ACE2 (protein–protein) interface were found mutated in different variants (Figure 2b), while a few substitutions at positions 446, 477, 478, 493, and 498 in the Spike RBD enhanced the affinity or binding of the RBD with the ACE2 receptor. In particular, regardless of the amino acids used for substitution, mutations at the 446 and 498 positions enhance ferritin–RBD affinity to the ACE2 receptor (Appendix A).

**Figure 2 vaccines-12-00446-f002:**
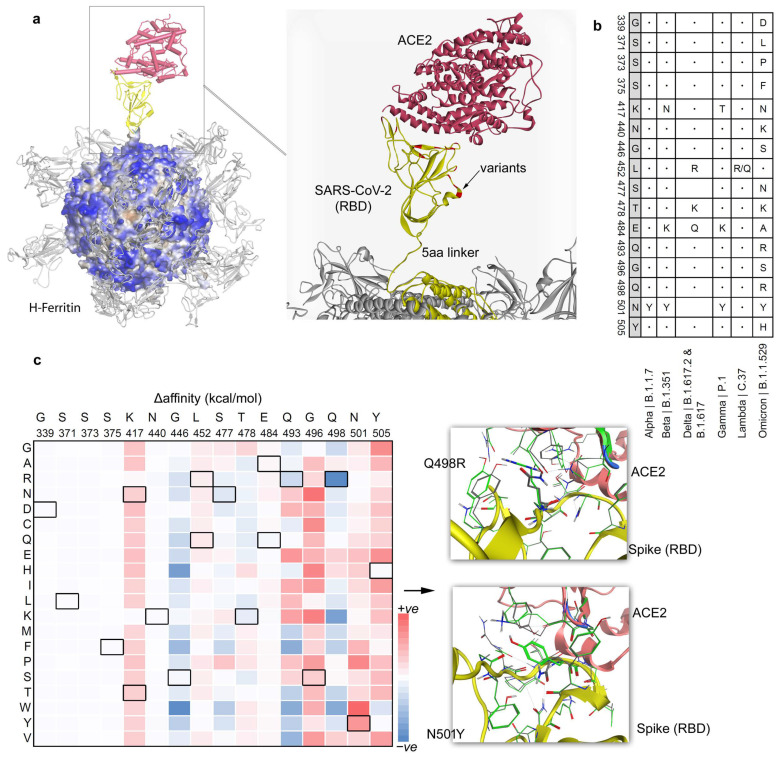
The SARS-CoV-2 Spike RBD–ACE2 interactions over the surface of the H-ferritin (24 subunits) nanocage. (**a**) The chimeric construct of the H-ferritin–GGGGS–Spike RBD investigated with the ACE2 host cell receptor. The right panel represents mutations derived from different SARS-CoV-2 variants within the Spike RBD domain (marked in red color): 24 subunits of Spike RBD (PDB ID: 6lzg [34]) presented over the H-ferritin (PDB ID: 2fha [47]). The outer and inner diameters of the H-ferritin cage are 12 nm and 8 nm, respectively. (**b**) Mutations from SARS-CoV-2 (Alpha, B.1.1.7), Beta (B.1.351), Delta (B.1.617.2 and B.1.617), Gamma (P.1), Lambda (C.37, and Omicron (B.1.1.529) [32,33] variants, residing within the RBD domain of the Spike protein. (**c**) Along with tracing change in the binding affinity upon inserting point mutation from different SARS-CoV-2 variants in the RBD domain with the ACE2 receptor, we investigated all different possible mutations at a particular site or residue. The heatmap represents change in the binding affinity (kcal/mol) between the Spike RBD and ACE2 receptor upon point mutations (A, R, N, D, C, Q, E, G, H, I, L, K, M, F, P, S, T, W, Y, and V). The right panel represents an example of a conformational switch of the residues upon mutation; Q498R (induced affinity) and N501Y (reduced affinity).

The effect of a particular mutation was performed by applying the ‘residue scan’ approach on the Spike RBD–ACE2 complex over the H-ferritin nanocage. The changes in the binding affinity between the Spike RBD–ACE2 and stability upon inserting point mutations were characterized (Figure 2 and Appendix A). Investigating the amino acid exchange found in selected mutations, K417N, K417T, L452R, L452Q, E484A, G496S, and N501Y have reduced the binding of Spike RBD to ACE2, whereas substitutions G446S, S477N, T478K, E484Q, Q493R, and Q498R have enhanced the binding affinity of Spike RBD–ACE2. A mutation-induced conformational switch was observed that could be responsible for increased (Q498R) or decreased (N501Y) binding affinity between Spike RBD and ACE2 (Figure 2 and Appendix A).

### 3.2. Binding of Spike RBD–Antibody Complex over Nanocages and Their Mutational Landscape

The Spike RBD, a very immunogenic and a major SARS-CoV neutralization determinant, can elicit effective neutralizing antibodies capable of outcompeting the ACE2 receptor. However, the structural basis of Spike RBD immunogenicity as well as Spike RBD-mediated neutralization remains limited [26]. We investigated different neutralizing antibodies such as VVH-72, B38, m396, CB6, S309, and CR3022 binding to the SARS-CoV-2 RBD domain over the H-ferritin nanocage constructs (Figure 3 and Appendix A). In addition, the ferritin–RBD–antibody complex was analyzed with the mutations derived from different SARS-CoV-2 variants (Alpha, B.1.1.7; Beta, B.1.351; Delta, B.1.617.2 and B.1.617; Gamma, P.1; Lambda, C.37; and Omicron, B.1.1.529). The mutations identified in different SARS-CoV-2 variants were explored by replacing a specific mutated residue with every possible 20 substitution using the ‘residue scan’ technique. Our findings revealed that site-specific mutations at a specific amino acid position can influence the Spike RBD-binding affinity with different antibodies over the H-ferritin nanocage (Figure 3 and Figure 4): VVH-72 (371, 373, and 375 Spike RBD amino acids; aa), B38 and CB6 (417, 477, 478, 484, 493, 496, 498, 501, and 505 aa), CR3022 (371 and 375 aa), S309 (339 and 440 aa), and M396 (375, 446, 493, 496, 498, 501, and 505 aa).

Mutations from different SARS-CoV-2 variants (Alpha, Beta, Delta, Gamma, and Omicron) were analyzed with VVH-72, B38, CB6, CR3066, m396, and S309 antibodies, and the mutation-induced changes in their binding affinity between Spike RBD were predicted (Figure 3c). In the ferritin–RBD–VVH-72 complex, the substitutions at S371L, N440K, and G446S positions have improved the binding energy, whereas mutations S375F and Y505H reduced the Spike RBD–antibody binding affinity. For the ferritin–RBD–B38 complex, the S477N, E484A, E484Q, E484K, and Q498R mutations gained the binding affinity, whereas K417N, K417T, L452R, L452Q, T478K, G496S, N501Y, and Y505H lowered the binding between Spike RBD and Ab (Figure 3c). Overall, the binding affinity of VVH-72, CR3022, and S309 antibodies with the Spike RBD demonstrated an unstirred effect upon most point mutations, although there were moderate binding affinities changes for a small sets of residues (Appendix A). The Spike RBD mutated in the presence of m396 (which binds to the RBD of SARS-CoV without functionally mimicking ACE2, preventing the RBD from binding to ACE2) and CB6 (functionally mimics ACE2 binding to the interface on RBD) has a higher influence in the protein–Ab binding affinities (Figure 4).

## 4. Discussion

Our findings and different studies revealed that the SARS-CoV-2 Spike RBD domain can be an attractive target for the development of RBD-based vaccines against different SARS-CoV-2 variants. In order to adapt distinct variants, a chimeric Spike RBD-dimer vaccine construct has been reported which stitches two RBD domains (Delta–Omicron or prototype–Beta conjugates) together originating from similar or distinct SARS-CoV-2 variants [44]. Implementing the surface plasmon resonance (SPR) assays, the binding of such Spike RBD-dimer vaccines with ACE2 was evaluated. Such chimeric-designed vaccine-treated mice showed an enhanced protection or neutralization of SARS-CoV-2 variants [44]. In addition, cryo-electron microscopy (cryo-EM) revealed that such Spike RBD-dimer vaccines, in particular prototype-Beta, can bind with CB6 mAb at 11.5–11.6 Å resolution [44]. Furthermore, the genetic fusion of the Spike RBD domains stitched over the ferritin nanocages subunits, making synthetic chimeras, may generate moderate to high levels of immune responses (Figure 2 and Figure 3), similar to that described in several studies [41,42,43,44,45,46].

The SARS-CoV-2 Spike expressed over the ferritin nanocages surface and their binding with different monoclonal antibodies were evaluated and demonstrated by the cryo-EM technique [41]. A single-dose vaccination of Spike presented over the ferritin nanocages enhances the stimulation of neutralizing antibodies, and a twofold increase (>convalescent plasma) in neutralizing antibody titers was reported upon single immunization with Spike–ferritin particles to mice [41]. Expressing the Spike RBD domain over ferritin nanocages in the eukaryotic expression system (293i cells) and immunizing rhesus monkeys (20 μg) with such constructs (Spike RBD–ferritin) induced the humoral immunity and T-cell response. In addition, cross-neutralizing antibodies have been reported against different SARS-CoV-2 strains [42]. Over the self-assembled ferritin nanocages with the Spike RBD domain, the Spike RBD-HR (heptad repeat) subunit was reported to be displayed that enhances the immune responses (B and T cells) along with the production of neutralizing antibodies [45]. In such a promising vaccination approach, the Spike RBD-specific antibodies blocked the interaction with the ACE2 host cell receptor [45].

Additionally, implementing the SpyTag-SpyCatcher system, different strategies have been applied to develop Spike RBD-based vaccine candidates, where the Spike RBD domain is conjugated with 24-mer ferritin, 60-mer mi3, and 120-mer I53–50 nanoparticles [46]. Immunizing mice with such a Spike RBD displayed over different nanoparticles induced a 8–120 fold greater neutralizing activity compared to monomeric Spike RBD treatment [46]. In addition, the mice immunized sera revealed that such Spike RBD-conjected nanoparticles can inhibit the binding activity of the Spike RBD domain with the ACE2 receptor [46]. Along with the expression of the Spike RBD domain over ferritin, the Spike protein has been found to be present over the Dps nanoparticles, and the RBD-S-Dps construct induced a neutralizing antibody response in mice [43]. The immunization of a single round with such a Spike RBD-S-Dps treatment to mice resulted in a reduced level of the virus in the lungs upon infection [43]. Moreover, combining different approaches (gene fusion, chemical conjugation, and SpyTag/SpyCatcher), nanoparticles were extracted from the turnip mosaic virus (TuMV; a crucial tool for diagnostics and immunodetection) that could represent a functional Spike RBD and enhance antibody sensing [48]. Ferritin nanoparticles have been reported to induce RFP (red fluorescence protein)-specific cytotoxic CD8+ T cell response as well as in live mice inhibiting tumor (melanoma) growth [39]. Falvo et al. reported the combination of ferritin nanoparticles stitched together with the antibody drug conjugate can enhance therapeutic indexes in melanoma tumors [40].

In our studies, we made use of the MD simulations, which can help to understand the structural folding or properties of a gene (structural or nonstructural) at an atomic scale along with predicting the possible variability that can emerge from a viral protein. Such techniques allow investigating model systems prior to validation along with selecting the most feasible domains of the target protein or a linker to be presented over the ferritin surface [16,49]. Intermolecular interactions between proteins over the simulation time from MDS can be considered to design SARS-CoV-2 S self-derived peptides [38] emerging within the viral components, and these can be displayed over the ferritin nanocages as potential blockers or inhibitors of the target protein. Considering the existing knowledge from the literature, the modeling techniques can predict the possible epitopes of new emerging viruses that can dock with the host cell receptors or ribosomal components [50]. Herein, our study we implemented MD simulation to find the most suitable linker for the ferritin–RBD constructs, and proposed that it may have the potential to induce a cross-protective antibody response. Such Spike RBD-targeting vaccines and neutralizing antibodies can effectively block the viral entry by interrupting the binding between Spike–ACE2 networks. Since this could provide useful therapeutic approaches against the current COVID-19 pandemic as well as future such infections, we optimized the structure of the H-ferritin–GGGGS–RBD complex which was further evaluated with the ACE2 host cell receptor or antibodies (Figure 1, Figure 2 and Figure 3).

## 5. Conclusions

In the field of nanomaterials, great progress has been reported, and ferritin nanocages with monodisperse architectures have been widely used in a number of disciplines, including vaccine development and generation. In our current work, we demonstrated the structural dynamics of ferritin nanocages stitched together with synthetic chimeras displaying the Spike RBD domains in the ‘up’ active conformations. Based on our findings, we postulated that such nanocage constructs may be an effective tool to induce cross-protective antibody responses. Taking the H-ferritin nanocage constructs into consideration, we evaluated the effects of variants on the binding of the SARS-CoV-2 Spike RBD to the host receptor ACE2 and different neutralizing antibodies. Our data predicted that these Spike RBD-targeting ferritin-based vaccines and neutralizing antibodies shall effectively block the viral entry by interrupting the binding between Spike and ACE2. Our in silico findings can, however, be validated by characterizing the morphology of the Spike RBD–ferritin nanocages and performing biochemical assays to evaluate their binding to ACE2.

Variants of concern have rapidly changed the transmissibility and pathogenicity of the SARS-CoV-2 virus and occurred mainly due to mutations in the main structural proteins. Variants within the Spike RBD regions lead or have been correlated to the emergence of novel SARS-CoV-2 variants such as Alpha (B.1.1.7), Beta (B.1.351), Delta (B.1.617.2 and B.1.617), Gamma (P.1), Lambda (C.37), and Omicron (B.1.1.529). The Spike RBD domain in the SARS-CoV-2 Omicron variant contains the highest number of mutations compared to other variants (α, β, γ, and δ). The effect of a particular mutation was performed by applying the ‘residue scan’ approach on the Spike RBD–ACE2/antibody complex over the H-ferritin nanocage. Changes in the binding affinity or structural stability between the Spike RBD antibody and stability upon inserting point mutations were characterized. Investigating the VOCs revealed that most of the variants reported a clearly reduced stability of the Spike RBD domain. Mutations have a moderate effect over the binding affinity for VVH-72, CR3022, and S309 Abs with the Spike RBD, whereas for the B38, CB6 or m396 Abs, they had a significant influence when evaluated over the H-ferritin nanocages. In particular, some mutations that induced conformational switches were observed that could be responsible for increased (Q498R) or decreased (N501Y) binding affinity between Spike RBD and ACE2. We believe these structural details can provide useful therapeutic approaches against SARS-CoV-2. With the undergoing grand challenges of designing vaccines against the current mutating SARS-CoV-2 and any future virus infections, we propose possible protein-based vaccine strategies along with evaluating their VOC profile.

## Figures and Tables

**Figure 1 vaccines-12-00446-f001:**
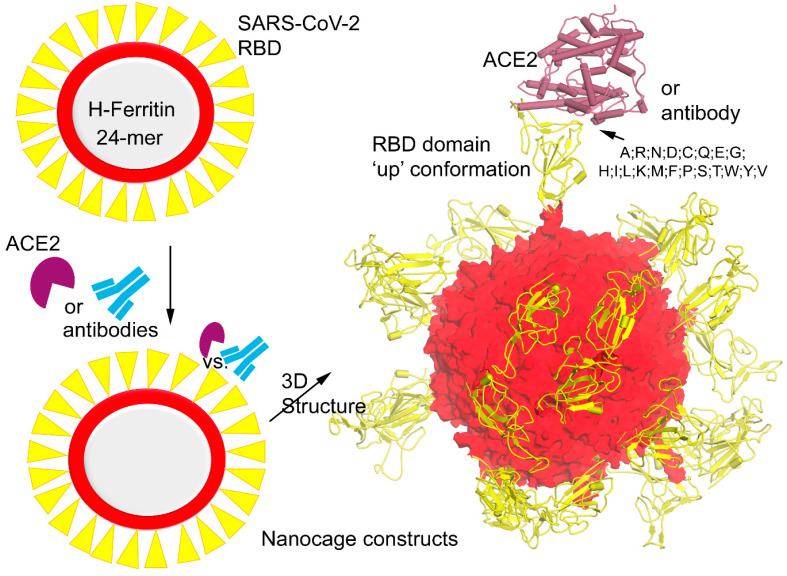
Diagram representing the model system of H-ferritin nanocage with SARS-CoV-2 Spike receptor binding domains (RBDs) docked with angiotensin-converting enzyme 2 (ACE2) host cell receptor and different antibodies. The right panel demonstrates the 3D structure of the Spike RDB domain in the ‘up’ active conformation [16] stitched together with the ferritin nanocage, and these protein–protein or antibody (Abs) interfaces were considered to investigate the mutational landscape over the Spike RBD domain.

**Figure 3 vaccines-12-00446-f003:**
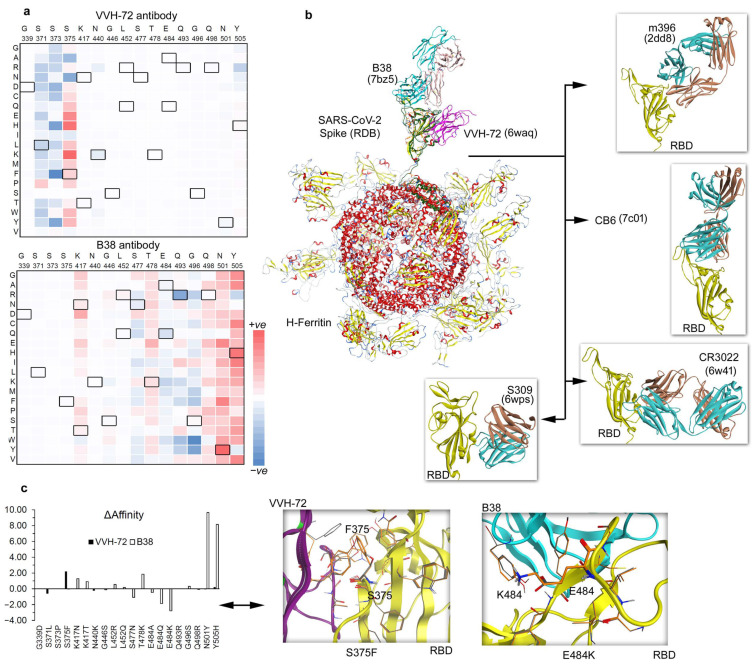
Ferritin nanocage constructs of the SARS-CoV-2 RBD domain with different antibodies. (**a**) Individual VVH-72 and B38 antibodies were built over the H-ferritin nanocage binding the Spike RBD domain, and the changes in the binding affinity were traced upon inserting point mutations. Mutations from SARS-CoV-2 virus variants (Alpha, B.1.1.7; Beta, B.1.351; Delta, B.1.617.2 and B.1.617; Gamma, P.1; Lambda, C.37; and Omicron, B.1.1.529) were further investigated by 20 different possible substitutions using the ‘residue scan’ approach. The heatmap represents a change in the binding affinity upon inserting mutations in the Spike RBD domains over the ferritin nanocage. (**b**) Investigating the antibodies binding with the SARS-CoV-2 RBD domain over the H-ferritin nanocage constructs. The VVH-72 (PDB ID: 6waq [30]) and B38 (PDB ID: 7bz5 [36]) imitates ACE2 binding to the interface on the RBD of SARS-CoV, m396 binds to the RBD of SARS-CoV without functionally mimicking ACE2, preventing the RBD from binding to ACE2 (PDB ID: 2dd8 [26]), CB6 functionally mimics ACE2 binding to the interface on the RBD of SARS-CoV-2 (PDB ID: 7c01 [27]), and S309 binds to the RBD of SARS-CoV-2 without blocking the binding of RBD to ACE2 (PDB ID: 6wps) [24,28]. In addition, the SARS-CoV specific CR3022 binds to the RBD of SARS-CoV-2 without blocking the binding of RBD of SARS-CoV-2 to ACE2 (PDB ID: 6w41) [24,29]. (**c**) The change in the binding affinity within the Spike RBD–antibody complexes upon inserting SARS-CoV-2 virus variants in the RBD domain. Individual effects of mutations describe a contrary dynamic in the presence of VVH-72 or B38. The left panel represents the change in the binding affinity within the Spike RBD antibodies upon inserting mutations in the SARS-CoV-2 RBD domain. The right panel describes the conformational switch of the S375F and E484K mutations with VVH-72 and B38 antibody, respectively.

**Figure 4 vaccines-12-00446-f004:**
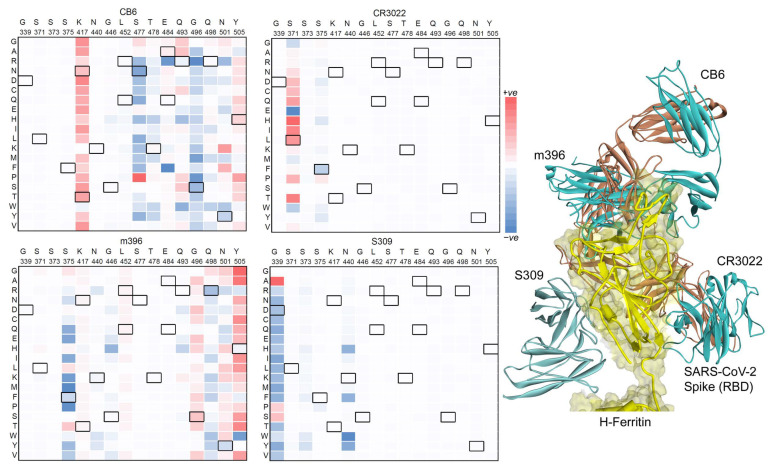
Investigating the CB6, CR3066, m396, and S309 antibodies binding with the SARS-CoV-2 Spike RBD over the H-ferritin nanocage. The left panel represents the heatmap of the affinity change upon the insertion of point mutations over the Spike RBD domain. The right panel describes the positioning or the location of all four antibodies over the ferritin–Spike RBD complex.

## Data Availability

Data is contained within the article or Appendix A.

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
