# Peer review of "Single Ferritin Nanocages Expressing SARS-CoV-2 Spike Variants to Receptor and Antibodies"

_vaccines, 2024, doi:10.3390/vaccines12050446_

Round 1
Reviewer 1 Report
Comments and Suggestions for Authors
In this study, Padariya et al assessed the structural dynamic of ferritin nanocages stitched together with synthetic chimera's displaying the SARS-CoV-2 Spike receptor binding domain (RBD) for their capacity to induce cross-protective antibody response. They used nanocage constructs designed in a previous work to evaluate in silico the effects of variants on the binding of SARS-CoV-2 Spike RBD to the host receptor angiotensin-converting enzyme 2 and different neutralizing antibodies. Their results suggested that most of the variant reported possibly enhanced stability of the RBD domain.
This is an interesting study concerning virtual screening of single ferritin nanocages expressing SARS-CoV-2 spike variants as an attractive molecule for vaccine development. The feasibility of this preventive approach and the applicability to in vivo models are not discussed.
Reviewer 2 Report
Comments and Suggestions for Authors
The manuscript extensively discusses how amino acid mutations in the RBD domain can cleverly reduce binding to the ACE2 receptor, thereby lowering viral infection, while hardly affecting the binding efficiency to antibodies VVH-72, CR3022, 244, and S309 Abs. This provides an innovative perspective for the design of nanovaccines.
1.However, what are the innovations and highlights of this manuscript compared to marketed RBD vaccines and RBD nanovaccines? In comparison to these works, what guidance can this study provide? Maybe you could refer to the following articles in the text:
(a)Marketed dimeric RBD recombinant protein vaccine,
(DOI: 10.1016/j.cell.2022.04.029).
(b)Articles reporting the preparation of RBD vaccines using ferritin nanocages and validating therapeutic effects in animal experiments,
(DOI: 10.1016/j.immuni.2020.11.015 and DOI: 10.1021/acsnano.0c08379).
2.Please ensure consistent chemical formula formatting:
(a)The single quote format differs in lines 77 compared to lines 43, 61, and 82; please ensure consistency.
(b)In line 147, ensure "pdb id: 2fha" is correctly formatted without missing the colon symbol ":".
3.Avoid repetitive explanations of the “MOE” abbreviation in lines 95 and 103; it should only be explained upon its first occurrence.
4. It is desirable to have experiments to characterize the morphology of the nanocages, which can be supplemented by experiments to verify the results of molecular dynamics simulations.
Comments on the Quality of English Language
There is some repetition of content between different sections of this manuscript, so it would be advisable to modify the wording and phrasing.
Reviewer 3 Report
Comments and Suggestions for Authors
Although this was hard work, the authors should improve their analyses and discussion concerning the following points:
1. Since rational protein design is more art than science such simulations must always verified by a biological assay. Therefore, the authors have to perform a proof of concept, e.g. by a MST measurement of the binding constants.
2. The simulations were performed with known knowledge of the binding of SARS-CoV-2 Spike RBD to the host receptor ACE2. The authors should discuss the forecast power of such simulations regarding future pandemics with new viruses and unknown host-pathogen interactions.
Reviewer 4 Report
Comments and Suggestions for Authors
Review of Manuscript “Single Ferritin Nanocages Expressing SARS-CoV-2 Spike 2 Variants to Receptor and Antibodies“ by Monikaben Padariya et al..
By in silico analysis the authors analyze the effects of point mutations in the receptor binding domain (RBD) of SARS-CoV-2 on RBD stability, the binding affinity to the ACE2 (angiotensin converting enzyme 2) receptor and the binding of neutralizing antibodies in the context of chimeras of RBD with ferritin nanocages. This is an interesting approach and generally the analyses are presented in a well ordered and comprehensible fashion. However, in the interpretation of their data, the authors should keep in mind that they exclusively have presented in silico data so far and that no additional data on in vitro or in vivo testing of the corresponding ferritin nanocage SARS-CoV-2 chimeras is included in the manucript. Thus in my opinion, statements as those in the abstract (lines 11/12, “…can be effective to induce cross-protective antibody response“ or lines 14 to 16, “can effectively block the viral entry by interrupting the binding between Spike-ACE2), but also in other parts of the manuscript, are a clear over-interpretation of the in silico data at the moment. This issue should also be addressed in the Conclusions section instead of giving mostly a mere repetition of the results.
As an additional major point of criticism, the linguistic design of the text can be greatly improved. Whereas the intention of most statements became clear for me as a reader with the corresponding scientific background, many formulations should be made more precise and the complete manuscript should best be corrected by a native English speaking person.
In addition to these general point, the major and minor issues listed in detail below should be addressed in a revised version of the manuscript.
Major points:
1) In lines 140/141 the authors report that “Accordingly, the substitutions at positions 446, 477, 478, 490, 493, and 498 in Spike RBD enhance the affinity or binding of the RBD with the ACE2 receptor“. However, in fig. S1a (right part), a mostly negative value (indicating an enhanced affinity) regardless of the amino acid used for substitution is only observed for the positions 446 and 498. For the other positions (477, 478,490 and 493), the mean effect of the different substitutions is around the zero mark. Furthermore, the term “accordingly“ somehow implies that the effects on binding affinity to ACE2 correlate with the effects on RBD stability. This is clearly not the case (fig. S1a, right panels).
2) Lines 162 to 166: “The effect of a particular mutation was performed by applying the ‘residue scan’ approach on the Spike RBD-antibody complex over the H-ferritin nanocage. Change in the binding affinity between Spike RBD-antibody and stability upon inserting point mutations were characterized (Figures 2 and 165 S1)“. Should this not read binding affinity of Spike RBD to the ACE2 receptor for this section and corresponding figures?
Lines 173 to 175: “The Spike RBD-antibody-Ferritin complex was analyzed with different SARS-CoV-2 variants (Alpha; B.1.1.7, Beta; B.1.351, Delta; B.1.617.2 & B.1.617, Gamma; P.1, Lambda; C.37, and Omicron; B.1.1.529“.
Please rather change to “…mutations from different SARS-CoV-2 variants…“, as variants with more than one point mutation were not really included in the analysis. This also applies to the statement in lines 203 to 205 and the legend to fig. 3.
Minor points:
Is there a special reason why the amino acids mutations at position 490 are shown included in fig. S1, but not in fig. 2?
Comments on the Quality of English Language
The linguistic design of the text can be greatly improved. Whereas the intention of most statements became clear for me as a reader with the corresponding scientific background, many formulations should be made more precise and the complete manuscript should best be corrected by a native English speaking person.
Round 2
Reviewer 1 Report
Comments and Suggestions for Authors
The authors have adequately responded to the issues raised and have revised the manuscript accordingly.
Reviewer 4 Report
Comments and Suggestions for Authors
Review of revised version of Manuscript “Single Ferritin Nanocages Expressing SARS-CoV-2 Spike 2 Variants to Receptor and Antibodies“ by Monikaben Padariya et al..
In the revised version of their manuscript the authors have tried to address most of the issues raised in my review of the original version. Regarding my questions of the implications of the in silico data for the binding between the SARS-CoV-2 receptor binding domain (RBD) and the ACE receptor in vitro and the potential neutralization of this binding through vaccination in vivo, the authors have addressed this issue in a detailed and competent manner in the newly introduced discussion section. Most of the other major points I raised were also addressed in a satisfactory fashion. The linguistic design has clearly improved, but there is still some scope for further improvements. I have listed some examples below under minor points without claim for completeness.
Minor points:
1) Line 109: References 37 to 44 should probably read references 39 to 44, as from the titles the references 37 and 38 are not related to the topic of blocking SARS-CoV-2 entry by antibodies. Also applies to line 234.
2) Line 125 typo: Change ‘thr’ to ‘the’.
3) Line 132: Change ‘reducing’ to ‘reduce’.
4) Line 138: Insert ‘, mutations’ after substitution.
5) Line 153: Change ‘mutated’ to ‘mutation’.
6) Line 155: Change ‘different’ to ‘selected’.
7) Line 157: Change ‘whereby’ to ‘whereas’ and omit the comma.
8) Line 235: Modify sentence ‘The cryo-EM data demonstrated the SARS-CoV-2 expressed over the ferritin nanocages along with their binding with monoclonal antibodies’ for a clearer statement.
9) Line 241: Change ‘of’ to ‘with’.
10) Line 260: Change ‘inducing’ to ‘induce’.
11) Line 308/309: Change ‘majority reducing’ to ‘clearly reduced’.
12) Line 316: Change ‘proposed’ to ‘propose’.
Comments on the Quality of English Language
Further improvement of English language required.
